# Recent Developments and Current Applications of Organic Nanomaterials in Cartilage Repair

**DOI:** 10.3390/bioengineering9080390

**Published:** 2022-08-15

**Authors:** Zhanqi Wei, Ganlin Zhang, Qing Cao, Tianhao Zhao, Yixin Bian, Wei Zhu, Xisheng Weng

**Affiliations:** 1Department of Orthopaedics, Peking Union Medical College Hospital, Chinese Academy of Medical Sciences and Peking Union Medical College, Beijing 100730, China; 2School of Medicine, Tsinghua University, Haidian District, Beijing 100084, China; 3Plastic Surgery Hospital, Chinese Academy of Medical Sciences and Peking Union Medical College, Beijing 100144, China; 4State Key Laboratory of Complex Severe and Rare Diseases, Peking Union Medical College Hospital, Chinese Academy of Medical Science and Peking Union Medical College, Beijing 100730, China

**Keywords:** organic nanomaterials, cartilage tissue engineering, regenerative medicine

## Abstract

Regeneration of cartilage is difficult due to the unique microstructure, unique multizone organization, and avascular nature of cartilage tissue. The development of nanomaterials and nanofabrication technologies holds great promise for the repair and regeneration of injured or degenerated cartilage tissue. Nanomaterials have structural components smaller than 100 nm in at least one dimension and exhibit unique properties due to their nanoscale structure and high specific surface area. The unique properties of nanomaterials include, but are not limited to, increased chemical reactivity, mechanical strength, degradability, and biocompatibility. As an emerging nanomaterial, organic nanocomposites can mimic natural cartilage in terms of microstructure, physicochemical, mechanical, and biological properties. The integration of organic nanomaterials is expected to develop scaffolds that better mimic the extracellular matrix (ECM) environment of cartilage to enhance scaffold-cell interactions and improve the functionality of engineered tissue constructs. Next-generation hydrogel technology and bioprinting can be used not only for healing cartilage injury areas but also for extensive osteoarthritic degenerative changes within the joint. Although more challenges need to be solved before they can be translated into full-fledged commercial products, nano-organic composites remain very promising candidates for the future development of cartilage tissue engineering.

## 1. Introduction

Cartilage is a special structure on the surface of the bone within the joint, only 2–4 mm thick, elastic, and smooth. The friction on the cartilage surface is very low, so it is easy to be injured during sports, and the cartilage becomes thin and even ruptures and wears out, which is called cartilage injury [1]. Clinical data show that among patients with cartilage injury in China, cartilage injury is concentrated in the patellofemoral joint and the lateral compartment, and a few are located in the medial compartment [2]. Due to the lack of blood vessels, lymphatic and nerve distribution in the articular cartilage, poor nutrition supply after injury, and very limited self-repair ability, it causes pain, joint instability, joint stiffness, and other clinical symptoms, secondary to the occurrence of osteoarthritis, which is an important cause of limb dysfunction and disability, seriously affecting the quality of life of patients, wasting medical resources, and causing a lot of inconvenience to patients [3].

In the current treatment methods, if a patient has a slight cartilage injury, certain immobilization should be given after the injury and generally 4–6 weeks for recovery. If a patient is suffering from a severe cartilage injury that affects the patient’s limb activity and function, surgery is generally required, and they can recover 2 months after the surgery. If the full layer of cartilage is damaged, the operation of chondrocyte culture and transplantation is required, and the postoperative recovery time is 6–12 months. The long convalescence makes the treatment of cartilage injury urgently require new treatment methods [4].

Nanotechnology and nanomaterials have made rapid development in recent years and are widely used in the medical field, providing new ideas for the treatment of cartilage injuries [5]. Nanomaterials can be divided into organic nanomaterials and inorganic nanomaterials. The role of organic nanomaterials in clinical cartilage injury and the progress of related nanotechnology will be introduced in this review [6].

## 2. Sports Injuries of Articular Cartilage

Articular cartilage is a layer of articular cartilage, either hyaline cartilage or fibrocartilage, that covers the surface of the bones attached to the joint [7]. Articular cartilage has neither nerves nor blood vessels, and its nutrition is mainly supplied by synovial fluid and arterial branches around the synovial layer of the joint capsule [8]. Between these collagen fibers are scattered chondrocytes, consisting of flat to oval or round cells from superficial to deep, that maintain the normal metabolism of articular cartilage [9]. Chondrocytes, along with the extracellular matrix (ECM), are the major components of articular cartilage tissue [10]. The damage and repair of articular cartilage is mainly related to ECM, while hyaluronic acid is the main component of the extracellular matrix and is directly involved in the regulation of electrolyte exchange inside and outside the cells [11]; in addition, type II collagen is widely distributed in articular cartilage and cross-linked with a small amount of type I and type IV collagen to form a network and increase mechanical strength [12,13].

During exercise, the cartilage of the joint is subjected to tremendous stress. In fact, the ability to damage the knee joint during exercise is related to the amount of pressure on the knee joint [14]. Studies have shown that human articular cartilage can withstand pressures of 25 MPa without significant damage, and that pressures above 25 MPa can lead to bone cell death and cartilage fractures [15]. Prolonged exposure to such stresses can lead to deformation and relaxation of the articular cartilage beyond the tolerable range [16]. Studies in animal models have shown that prolonged high-pressure impacts on joints lead to a sustained release of tissue factors that induce inflammation, loss of tissue integrity and mechanical properties, and eventually cell death [17]. The main cause of articular cartilage damage is mechanical injury, which later further leads to secondary osteoarthritis [18]. Mechanical injury can either directly lead to the destruction of the tissue matrix or be mediated by chondrocyte expression of matrix-degrading enzymes or reduced matrix synthesis activity [19]. As a result, the mechanical properties of the cartilage will be significantly reduced.

## 3. Current Clinical Methods of Cartilage Repair

For the repair of articular cartilage defects, the strategy used depends on the degree of damage and degeneration [20]. Articular cartilage defects that do not penetrate the subchondral bone cannot be healed by the body’s own repair function because the bone marrow space is difficult for the body’s stem cells to access. In contrast, defects that penetrate the subchondral bone have been shown to be repaired by the body’s intrinsic repair processes [21]. After repair, fibrocartilage tissue is usually formed, which has inferior properties compared to hyaline cartilage. Defects in the articular cartilage are usually treated with tissue implantation or cartilage regeneration methods [22]. Currently, the treatment of osteochondral defects can be divided into nonsurgical and surgical treatments.

Nonsurgical treatment is divided into physical therapy and nonphysical therapy for cartilage injuries that are combined with osteoarthritis rather than isolated cartilage injuries. Physical therapy is the most common option for nonsurgical treatment, and the main tools include pulsed electromagnetic fields and low-intensity pulsed ultrasound. Physical therapy provides better pain relief and improved knee function than nonphysical therapy. Although physical therapy can provide symptomatic relief, it cannot treat cartilage damage and concurrent osteoarthritis [23]. The main nonphysical treatment tools are oral nonsteroidal anti-inflammatory drugs (NSAIDs) and intra-articular injections of chondroitin sulfate and glucosamine. The latter two are commercially available products derived from hyaline cartilage, both of which have been shown to have good effects on cartilage injury and osteoarthritis with a better prognosis. However, this method is relatively unreliable and lacks support from clinical data, and the criteria for evaluating the analgesic function of chondroitin sulfate or glucosamine are vague [24]. In addition to chondroitin sulfate and glucosamine, hyaluronic acid is also widely used. Hyaluronic acid is a major component of synovial fluid, relieves moderate pain, and contributes to chondrogenesis [25].

Bone marrow stimulation is used in surgical operations for cartilage reconstruction, the core technique of which is autologous chondrocyte implantation (ACI). ACI is more often used to treat extensive symptomatic defects by using nonosteoarthritic cartilage to wrap around the defect site, thereby artificially creating a cavity in the damaged area of the cartilage, and then depositing the implanted cells to produce new tissue [26]. Biodegradable three-dimensional matrices are now being rapidly promoted for use in periosteal patches, replacing the traditional technique of chondrocyte injection. The biodegradable 3D matrix not only has good biocompatibility to suit the nature of the repair site, but also accelerates the process of matrix healing [27]. The main components of this material are hyaluronic acid and type II collagen-based material [28]. In addition, several natural or synthetic materials have been developed by medical institutions that stimulate the microenvironment of joint tissue and enhance its mechanical structure [29]. However, the problems faced by surgery are slow recovery, high risk, and a maintenance time of only 15 years, which cannot be completed once and for all, and only play the role of relief. The effect is not obvious, if the repair with autologous cartilage requires two surgeries, and the tolerance of the tissue structure is very limited, but also easy to cause damage to other parts [4].

The latest clinical treatments include platelet-rich plasma therapy (PRP) and stem cell therapy. The former cannot only promote the repair of articular cartilage and meniscus damage, but also promote the absorption of inflammation in the knee joint, with high efficiency in pain relief [30]. The latter is currently at the stage of clinical translation and has not yet been promoted on a large scale, and its effectiveness and safety are still not guaranteed [31]. Moreover, Kim et al. [32] showed in vivo differentiation studies that the number of cytokines inducing mesenchymal stem cell (MSC) differentiation to chondrocytes in the body tissue is low, and only a small fraction can be transformed into chondrocytes.

In general, surgical treatments are invasive, induce stress reactions in the body, have a narrower scope of application, and cause more discomfort to the patient; nonsurgical treatments mostly have a longer treatment period and slower recovery, and are only applicable to mild articular cartilage injuries [5]. However, the development of organic nanomaterials is expected to overcome these drawbacks simultaneously.

## 4. Application and Advantages of Organic Nanomaterials in Cartilage Repair

In modern clinical treatment, tissue engineering is widely used to repair cartilage injuries. The core of tissue engineering repair is the use of nanomaterials to build scaffolds that provide a suitable 3D environment for stem cell regeneration and differentiation [33]. Over the past two decades, a range of nanomaterials has been used in a wide range of cartilage repair and regeneration processes [34]. Nanomaterials are divided into nano-organic materials and nano-inorganic materials, while nano-organic materials are divided into polysaccharide-based nanomaterials and protein-based nanomaterials [35].

A typical application of inorganic nanomaterials for cartilage repair is carbon nanotubes. With high mechanical strength, fatigue resistance, and ductility, carbon nanotubes have great advantages as scaffolds for de novo cartilage, but poor integration with cartilage often leads to implant failure; in addition, excessive elastic modulus leads to stress shielding, resulting in low cartilage density, making the overall mechanical strength of the tissue insufficient and often requiring further surgical intervention and hardware fixation [36]. Amiryaghoubi et al. [37] reported that the existing clinical experimental approach for carbon nanotubes relies on laborious arthroscopic interventions combined with bone marrow infusion, stem cell injection, or transplantation of different connective tissues. These techniques are still in the experimental phase, with mixed results and a few long-term follow-up studies.

Nano-organic materials are more suitable for tissue engineering because they effectively promote the infiltration, transfer, and proliferation of cartilage stem cells in terms of surface properties, such as topological structure and hydrophilic qualities, than traditional organic materials. The special structure of nano-organic materials not only provides a larger specific surface area for the activity of stem cells, but also provides a microenvironment with suitable ionic strength, and mimics the natural factors that promote cartilage repair in the in vivo environment, which have been confirmed in clinical studies [38]. A major advantage of nano-organic materials over conventional nano-inorganic materials is their excellent biocompatibility, which is a type of biomaterial that can be used to treat or replace damaged cartilage tissue [39]. Currently, most of the nano-organic materials are involved in cartilage repair in the form of 3D scaffolds [40]. To improve the quality of 3D scaffolds, nanotechnology, such as spinning, chemical etching, 3D printing, and phase separation, has been widely used [41]; nanoparticle technology has also been put into use to enhance the interactions between chondrocytes and 3D scaffolds [42]. These nanotechnologies, together with organic nanomaterials, have enabled the customization of different cartilage tissue structures according to clinical needs [43].

### 4.1. Nanomaterials Based on Polysaccharides

The main polysaccharide-based nanomaterials (i.e., polysaccharide-based composite nanomaterials composed of polysaccharides and multifunctional inorganic nanoparticles) commonly used in cartilage repair include chitosan, alginate, agarose, and hyaluronic acid. Each of these materials will be described later.

#### 4.1.1. Chitosan

Chitosan is the product of the deacetylation of N-chitin [44]. Chitin, chitosan, and cellulose are relatively similar in chemical structure; specifically, cellulose has a hydroxyl group at the C2 position, while chitin and chitosan are replaced with an acetylamino and an amino group at the C2 position, respectively [45]. The chemical structure determines the properties, so chitin and chitosan have many unique properties, such as biodegradability and cell affinity. In addition, chitosan containing a free amino group is the only basic polysaccharide among natural polysaccharides. In recent years, a large number of research results have been obtained on the application of chitosan-based materials to cartilage tissue repair [46]. Pace et al. [47] reported that this natural biomaterial has excellent biocompatibility, which not only reduces cytotoxicity, but also is easily processed into various geometries. The special nanostructure of chitosan allows cell growth and forms a scaffold for cartilage stem cell attachment [48]. For example, hydroxyapatite (HAp)/chitosan-pectin (nHCP) composites were synthesized with in situ mineralization reactions in chitosan-pectin polyelectrolyte complex (PEC) networks [49]. The pH of the microenvironment and the chitosan/pectin ratio during the synthesis process play an important role in the formation of nano-chitosan complex crystals [50]. The good cytocompatibility of chitosan is reflected in the fast gelation properties of its solution, a property that mimics some of the properties of ECM, providing a cartilage matrix for the accumulation and crawling of newborn chondrocytes [51]. Gels composed of chitosan can remain for more than a week at sites of damaged cartilage that have been fixed, and for at least a day at sites of cartilage defects that have not been completely fixed [52]. In addition, composites composed of chitosan and other materials also play a good role in cartilage repair. For example, the chitosan-ethylene terephthalate mesh scaffold plays an important role in the repair of ECM thickness and the regeneration of type II collagen at the site of cartilage defects [53]. Chitosan can also be mixed with polycaprolactone (PCL) solution to form a scaffold for cartilage repair, and depending on the proportion occupied by both, the solution has different properties to maximize the speed of cartilage repair, degree of deacetylation, and mechanical strength, respectively [54,55]. Chitosan hydrogels also have unique temperature-sensitive properties that maximize the repair and reconstruction effects of the hydrogel when the local temperature of the cartilage defect site is appropriate, which also suggests that we should use thermostatic devices in future clinical treatments [56]. For implants, chitosan-containing implants perform better, as evidenced by better biocompatibility leading to slower clot formation, inhibition of vasoconstrictor nerves, less release of interleukins and other inflammatory factors, and higher mechanical strength and elastic modulus detectable in chitosan-containing implants [57,58].

#### 4.1.2. Alginate

Alginate is a natural polysaccharide found in the cell walls of brown algae. Usually, the pure product is white to brownish yellow fibers, granules, or powder. Alginate readily forms gels with cations, such as sodium alginate, and is known as alginate [59]. Alginate is a polysaccharide formed by the linear polymerization of monoglycoalkalic acid, the monomers being β-D-mannuronic acid (M) and α-L-gulonic acid (G). The M and G units are linked by 1–4 glycosidic bonds as block copolymers in combinations of M-M, G-G, or M-G, with molecular weights ranging from 10,000 to 600,000 [60]. Alginate has an ideal degradation rate and a range of excellent properties, and alginate oxide (OA) is now commonly chosen clinically as an alternative to conventional implants and has been introduced into a range of technologies, such as hydrogels, microspheres, 3D printed/composite scaffolds, membranes, and electrostatic spinning and coating materials [61]. By utilizing OA, OA-based materials can be easily functionalized and delivered with drugs or growth factors to promote cartilage tissue regeneration [62,63]. The modern theory of “wet wound healing” proves that the healing process is accelerated when wounds are exposed to a moist environment [64]. As a natural polymer derived from brown algae, alginate is highly hygroscopic, hydrogel-forming, and has excellent biocompatibility. Alginate has been extensively investigated as a raw biomaterial for the manufacture of wound healing dressings [45]. So far, the main forms of alginate wound dressings are fibers, sponges, hydrocolloids, hydrogels, etc. In general, alginate fibers are the most widely used clinically among the four alginate-based wound healing dressings [65]. Nevertheless, alginate sponges, hydrocolloids, and hydrogels have received increasing attention in both basic and clinical research due to their excellent properties in promoting wound healing [66]. Alginate-based implanted scaffolds facilitate the growth of chondrocyte stem cell adsorption. Antich et al. [67] confirmed that HAp/alginate nanocomposite scaffolds have been prepared by electrostatic spinning, successfully mimicking the in-situ synthesis of type II collagen in cartilage tissue. This method enabled the uniform deposition of HAp nanocrystals on the lamellar coating, overcoming the severe agglomeration of HAp nanoparticles processed with conventional mechanical electrostatic spinning methods [68]. The special spatial structure of the nanosheets can bind HAp and interoperate with alginate to form composites, which are advantageous in applications of cartilage tissue repair [69]. In addition, repair of cartilage damage is now clinically accomplished using alginate hydrogels, and Baba et al. [70] used a combination of alginate microspheres and a porous polyvinyl alcohol hydrogel scaffold to successfully demonstrate the role of the new composite in managing mechanical specifications and enhancing cell migration and the feasibility of repairing cartilage defects after composite scaffold implantation. Liu et al. [71] showed that articular cartilage cells seeded in alginate hydrogels increased Young’s modulus and mechanical stiffness over time and significantly increased the initial hyaline cartilage biomass. Alginate hydrogels have a wide range of action, even human dental pulp stem cells can be regenerated in them, and almost all sites of chondrocytes do not cause an inflammatory response during regenerative repair in alginate hydrogel scaffolds, demonstrating the stabilizing effect of alginate hydrogels on inflammatory cells [72].

#### 4.1.3. Agarose

Agarose is an organic substance that is a white or yellow bead-like gel particle or powder at room temperature, a linear polymorph with a basic structure of long chains of β-D-galactose and 3,6-endoether-l-galactose alternately linked together [73]. Agar pectin is essentially a mixture of many kinds of small molecules. Agarose needs to be dissolved in water temperatures above 90 °C and forms a semi-solid gel when the water temperature drops to 35–40 °C, when the state is most stable [74]. This phase change property is the physical basis for the versatility of agarose in the field of cartilage repair. Gel strength is used to characterize agarose gel properties, with higher values indicating better gel properties, and the strength of the mechanical properties of the cartilage tissue to be repaired is directly related to the rate of ECM synthesis. Therefore, attention needs to be paid to promoting the synthesis and secretion of ECM in chondrocytes in clinical treatment, which contributes to the tissue engineering of cartilage scaffolds [75]. Garakani et al. [76] proved that mechanical stimulation is an effective method to enhance cartilage extracellular matrix synthesis, and agarose, which has both biocompatibility and some mechanical strength, has been widely used as a cell culture scaffold for mechanical stimulation studies. When chondrocytes are attached to a scaffold composed of agarose gel, the action of external forces can accelerate the directed differentiation of chondrocytes, while stimulating the proliferation of differentiated chondrocytes and the secretion of ECM [77,78]. Although the applied mechanical load can induce the differentiation of chondrocytes and also promote the synthesis of the extracellular matrix of cartilage tissue, it is not the only factor that affects the synthesis of cartilage matrix; the surface structure of the material that encases the chondrocytes is also crucial. The surface structure of the material encasing the chondrocytes is also crucial [79]. Salati et al. [73] reported that the surface structure of agarose-based compounds is particularly suitable for chondrocyte encapsulation. In addition to their encapsulation role, the production of chondrocyte-derived glycosaminoglycans (GAGs) highlights their important role in cartilage repair. Ateshian et al. [80] verified that the differentiated adipose-derived adult stem cells will synthesize and secrete proteoglycans, hydroxyproline and sulfated GAG (sGAG) after being embedded in alginate hydrogel and agarose culture for a period of time and induced using TGFβ-1. In addition, Schmidt et al. [81] demonstrated that the biological properties of agarose scaffolds are uniquely close to native articular cartilage, and the mechanical properties of cellular-agarose hydrogel scaffolds are comparable to those of natural articular cartilage. Felfel et al. [82] reported that encapsulation of immature articular chondrocytes in agarose hydrogels significantly increases the repair capacity, a quality that is superior to that of native cartilage. With this method, chondrocytes can recover rapidly in culture as long as the integrity of the tissue remains intact during development, significantly shortening the recovery period.

#### 4.1.4. Hyaluronic Acid

Hyaluronic acid is a glycosaminoglycan, formed by the polymerization of a double pond unit consisting of D-glucuronide and N-acetylglucosamine [83]. Hyaluronic acid is an acidic mucopolysaccharide with a variety of special properties, such as biocompatibility and macromolecular adhesion, which can reduce joint friction, adjust the permeability of blood vessel walls, adsorb and release proteins, assist in transmembrane transport of metal ions and anions, and accelerate the recovery of cartilage damage in vivo [84]. Osteoarthritis (OA), usually caused by rupture of the articular cartilage and the underlying bone tissue, is a common cartilage disease and accounts for the vast majority of sports-induced cartilage damage [85]. Li, Qi et al. [86] proved that injectable hydrogels of hyaluronic acid with Epigallocatechin-3-gallate (EGCG), which has intrinsic properties that modulate inflammation and scavenge free radicals, can control inflammation and enhance cartilage regeneration when injected into the joint cavity. Composite hydrogels were prepared by mixing EGCG, tyramine complexed HA, and gelatin together in appropriate proportions. The composite hydrogel can adsorb the proinflammatory factor IL-1β, thus, protecting the newborn chondrocytes and promoting the regeneration of cartilage in vitro. Additionally, in in vivo histological experiments, the EGCG-HA/gelatin hybrid hydrogel has a good effect on the repair of cartilage damage caused by a sports injury, which can greatly reduce the loss of cartilage [87]. Microgels, as a special form of a hydrogel, can achieve precise regulation of the cellular microenvironment at the microscopic scale, providing a new avenue for regenerative cartilage tissue repair [88]. Martin et al. [89] showed that microgels encapsulating mouse bone marrow mesenchymal stem cells (mBMSC) prepared by microfluidic technology with vinylsulfonated hyaluronic acid and mercapturic gelatin as the main materials have good functions in promoting cartilage repair. After a period of cartilage induction in vitro, the microgels were injected into nude mice, and Zhang et al. [90] confirmed that the microgels could spontaneously form cartilage-like self-assemblies due to intercellular interactions and secretion of extracellular matrix, and BMSC changed from hyaline to fibrochondrogenic differentiation with the increase in cross-linking degree. A commercial hyaluronic acid-based polymer (Hyaff-11) currently supports the ability of human bone marrow mesenchymal cells (hMSCs) to chondrogenic differentiation by inducing the upregulation of type II collagen, type IX collagen, and aggrecan, as well as a decrease in type I collagen expression. In addition, Wong et al. [91] reported that differentiation of hMSCs into chondrocytes is induced in the presence of higher concentrations of TGFβ-1. Overall, the advantages of hyaluronic acid-based biomaterials include water solubility, gelling ability through reduced temperature, and biological properties (e.g., noncytotoxicity and cytocompatibility) that make them suitable candidates for cartilage tissue repair [92]. For example, chitosan-hyaluronic acid dialdehyde hydrogels can induce bone marrow cell differentiation in vivo and further promote ECM proliferation in hyaline and fibrocartilage [93].

### 4.2. Nanomaterials Based on Protein

The main protein-based nanomaterials (i.e., composites based on protein macromolecules combined with inorganic or organic small molecules) commonly used in cartilage repair are collagen and fibrin. They play different roles in the repair of cartilage tissue based on their unique properties.

#### 4.2.1. Collagen

Collagen accounts for approximately 20% of the total mammalian protein and is a very important protein of the human ECM, mainly found in connective tissue. Collagen has a strong elongation capacity and is the main component of ligaments, and collagen is also a major component of the ECM [94]. Collagen microfibrils are the most basic components of collagen. Many collagen microfibrils accumulate laterally and are linked in the same way by covalent bonds to form collagen fibers. Collagen fibrils are the basic forms of collagen for its physiological functions, and in living organisms they are interwoven into a mechanically strong and elastic meshwork that becomes the most basic component of connective tissue [95]. Exercise causes progressive wear and tear of the articular cartilage, which leads to loss of cartilage tissue, resulting in increased exposure at the ends of the long bones, decreased protection of the epiphyseal cartilage, and finally degenerative osteoarthrosis [96]. The regenerative capacity of cartilage tissue is poor, and cartilage healing is relatively more difficult after sports injuries [97]. In current clinical practice, three-dimensional (3D) porous scaffolds filled with cartilage stem cells are widely used for cartilage tissue repair, with relatively good repair results. However, most of the scaffolds currently use organic solvents to cross-link small molecules or use chemically synthesized polymers [98]. Lee and Kim [99] demonstrated that collagen and oxidized hyaluronic acid-based composite scaffolds have high biocompatibility and excellent mechanical properties, and can promote angiogenesis and chondrocyte proliferation; if metal ions are added to this scaffold, the mechanical strength of the scaffold will be close to that of native cartilage, and will not cause inflammation [100]. The cartilage bionic matrix material prepared from type II collagen is a gel at body temperature and a sol-gel at low temperatures, which can anchor cartilage stem cells to the damaged area by phase change [101]. The method is simple and easy to use, the product is easy to use, the surgical damage is small, the defect site is repaired well, and the clinical application is safer [102].

#### 4.2.2. Fibrin

Fibrin is a fibrous, nonspherical protein involved in blood clotting. It is turned into fibrin monomers by the action of thrombin on fibrinogen, which then polymerizes to form fibrin [103]. Fibrin is required in the following biological processes: messaging, blood coagulation, platelet activation, and protein polymerization. Human fibrin gel is a Food and Drug Administration (FDA)-approved material that mimics the coagulation process and can be used as a matrix for cartilage tissue engineering [104]. Platelet-rich fibrin is a platelet polymer tightly encapsulated by fibrin, with a loose and porous interior, and contains a variety of cell growth factors that promote chondrocyte differentiation and proliferation [105]. Platelet-rich fibronectin not only has anti-infective and anti-inflammatory effects, but also promotes tissue healing and regeneration, and contains a variety of cytokines that promote the differentiation of bone marrow MSCs into chondrocytes [106]. Platelet-rich fibrin can repair the old cartilage defect caused by traumatic osteoarthritis, and its repair ability is significantly better than that of BMSCs alone. The old cartilage defect is one of the common types of cartilage injury caused by exercise [107]. Masgutov et al. [108,109] reported that the fibrin glue cell complex by adding phalloidin and tranexamic acid, and they found that the modified cells generated new cartilage with histological properties consistent with normal cartilage and accelerated chondrocyte ECM formation and cartilage repair. Currently, Vilar et al. [110,111] have injected fibronectin sealant (FS) into the joint to form a trap similar to normal cartilage tissue, in which new chondrocytes are located to secrete vigorously and build injectable cartilage tissue together with FS. In addition, Heo et al. [112,113,114] have demonstrated that fibrin can also be involved in constituting nanohydrogels, which stimulate GAG production and ECM formation and can improve the mechanical strength of new cartilage and accelerate the delivery of adipose-derived multipotential stem/progenitor cells (ASPCs) to the damaged tissue for continued differentiation to form new cartilage tissue.

### 4.3. Nanosynthetic Materials

Natural cartilage has a complex layering in which the ECM is rich in nanoscale collagen fibrils and proteoglycan molecules, which provide many nanostructural features to natural cartilage tissue. Therefore, if one hopes to repair and regenerate cartilage using organic nanomaterials, then this material needs to have both excellent mechanical properties and good biocompatibility to mimic the nanostructural features of natural cartilage [115]. These nanomaterials are polymers, synthesized artificially, and differ significantly from the previously described natural nano-organic materials in terms of their scope of application and mechanism of action. Different types of conventional materials are used in this field, namely polylactic acid (PLA) and its derivatives poly-L-lactic acid (PLLA), poly (lactic-co-glycolic acid) (PLGA), dextro-polylactic acid (PDLA), polyurethane (PU), polyethylene glycol (PEG), and polyvinyl alcohol (PVA). In addition to their good mechanical properties, they also have good processing potential [116]. Additionally, similar to natural organic nanomaterials, synthetic nanomaterials are widely used for the preparation of hydrogels [117]. Such hydrogels show high efficiency in immobilizing living cells, including cartilage stem cells, and creating a highly hydrated microenvironment that allows easy diffusion of nutrients and induces cell migration, proliferation, and differentiation [118].

#### 4.3.1. PEG

PEG has been extensively studied as a support agent for cartilage tissue engineering. The main application directions of PEG are involved in the synthesis of hydrogels and for 3D bioprinting [119]. The hydrogel scaffold composed with the involvement of polyethylene glycol can promote the cell viability of cartilage stem cells and contribute to the attachment and growth of new cells and the generation of ECM [120]. When polyethylene glycol and filamentous protein are configured together as inks for 3D bioprinting, the printed synthetic cartilage scaffolds will promote differentiation of MSCs and secretion of ECM, and have phase change properties to facilitate implantation into damaged areas [121].

#### 4.3.2. PVA

Polyvinyl alcohol (PVA) is a synthetic polymer that is soluble in water and forms a solution with special adhesive properties. The materials involved in the composition of PVA mostly have superior properties for adhering to living cells, which can speed up the repair process and hold promise to be widely used in cartilage repair [122]. The mechanical strength of 3D scaffolds made with PVA is similar to that of natural cartilage, and the PVA-based hydrogel has a better tensile modulus than natural cartilage and is biocompatible, which can repair articular cartilage defects and delay the onset of degenerative changes after implantation in joints [123,124]. PVA mixed with other nanomaterials can also have good effects, for example, when mixed with titanium fiber mesh, it can help the growth of subchondral bone and play a role in fixation [121]; the composite material mixed with chitosan can promote the proliferation of bone marrow mesenchymal stem cells to assist in the repair of cartilage defects [125].

#### 4.3.3. PLGA

PLGA has been widely used in various medical engineering applications because of its good biocompatibility and degradability, and the special feature of the 3D scaffold created with PLGA is that it has relatively larger pores, which helps cells to infiltrate and migrate along the scaffold without being adhered to the scaffold, accelerating the regeneration of the cartilage defect site [126]. Kim et al. [127] showed that the composite scaffold prepared by PLGA and decellularized articular cartilage extracellular matrix (DACECM) is noncytotoxic and has mechanical strength and elasticity comparable to that of native cartilage. Qu et al. [128] also mixed PLGA with type II collagen and GAG to prepare scaffolds and found that they possessed superior compression modulus without significant changes in other properties, making PLGA more suitable for cartilage repair. As shown in Figure 1, Shen et al. [56] incorporated PLGA short fibers into a chitosan hydrogel scaffold for mechanical strengthening and structural biomimicking; meanwhile introducing cartilage-decellularized matrix (CDM) for biochemical signaling to promote chondroinduction activities. They found that the incorporation of PLGA short fibers and CDM remarkably strengthened the mechanical properties of the chitosan hydrogel. Biologically, the scaffold significantly promoted the adhesion and proliferation of chondrocytes and supported the formation of matured cartilage tissue with a cartilage-like structure and the deposition of abundant cartilage ECM-specific GAGs and type II collagen. They thereby demonstrated the great potential of PLGA in cartilage tissue repair and regeneration.

#### 4.3.4. PCL

PCL is widely used as a medical biodegradable material because of its good biodegradability, biocompatibility, and nontoxicity. PCL is used in electrospinning technology because it can be complexed with metal anions, and the composed nanoscaffold can induce differentiation of mesenchymal stem cells and assist in the formation of cartilage matrix [129]. In addition, compared with conventional scaffolds, PCL scaffolds have smaller pores, so the mechanical strength is higher, even higher than that of natural cartilage, but the small pores also prevent cell penetration, so uneven distribution of chondrocytes may occur [130,131].

#### 4.3.5. PLLA

PLLA is an important biodegradable polymer material that is characterized by nontoxicity, nonirritation, biodegradable absorption, high strength, good plasticity, and easy processing and molding [132]. Zhao et al. [133] prepared PLLA as nanofibrous scaffolds and found that the cells penetrated more uniformly; the surface of new cartilage was smoother, and more type II collagen was secreted, resulting in better mechanical properties. It has been reported that when PLLA scaffolds were implanted, the scaffolds themselves were completely resorbed after a period of time, but the new chondrocytes and the secreted ECM retained the shape of the scaffolds, which facilitated subsequent defect repair, and the new cell population was stable in nature [134]. Additionally, Mahboudi et al. [135] reported that the type of collagen and the amount of GAG were relatively higher in the new cartilage tissue repaired by PLLA scaffolds, while the number of macrophages was lower and the bioaffinity was higher.

#### 4.3.6. PU

One of the major challenges in cartilage repair is the integration of new cartilage with the original bone tissue, and the nature of PU can better solve this problem [136]. Abpeikar, Wen et al. [137,138] verified that the cartilage repair scaffolds prepared with PU can help stem cells replicate in situ and have good mechanical properties; more importantly, PU can be used to prepare scaffolds using a gas foaming method [139], and the resulting porous scaffolds have high cell adhesion and elasticity, which can effectively bind growth factors [140,141]. It can create a more stable microenvironment that is more conducive to cell adhesion and proliferation [142,143].

### 4.4. Next Generation Organic Nanomaterials

The organic nanomaterials described in detail in the previous section, i.e., agarose, alginate, hyaluronic acid, collagen, PLA, PVA, etc., are considered to be classical materials for articular cartilage repair. Adjusting the ratio of different base materials in composites can yield composites with different mechanical and biological properties, but qualitative changes in material properties are difficult to occur. With the advances in materials science and biomedical engineering, the development of nano-biomaterials for cartilage tissue engineering applications requires an integrated consideration of the interactions between polymer science, nanoscience and cell biology, based on which next-generation organic nanomaterials promising for cartilage repair have been developed in the laboratory and will be described in the following sections.

#### 4.4.1. Double Network (DN) Hydrogel

Hydrogels show potential for a wide range of applications in the field of tissue engineering. The field of cartilage repair places higher demands on hydrogels that can withstand continuous high levels of impact over short periods of time while having self-healing properties after damage [144]. Unfortunately, the inherent structural inhomogeneity of hydrogels makes the mechanical strength of conventional hydrogels relatively low and does not fully mimic the growth process of cartilage tissues [145]. To overcome this problem, attempts have been made to strengthen the mechanical strength of hydrogels by means of composite materials and multiple architectures, such as DN hydrogels [119]. The steps for preparing DN hydrogel are shown in Figure 2. DN hydrogels not only have better biocompatibility compared with previous generation hydrogel materials to accelerate the regeneration of hyaline cartilage in vivo, but also have higher material science properties, such as mechanical strength, toughness, corrosion resistance, as well as fatigue resistance, self-recovery shape memory, pH-mediated phase change, thermoplasticity (which can be used for 3D printing), and a series of engineering properties, thus, comprehensively surpassing single-network hydrogels [146]. The bilayer network structure of DN hydrogels allows it to combine the properties of different materials [147]. Wang et al. [148] used biomacromolecular polymers to form the first network, which helps complete the enzymatic reaction, and biocompatible polymers to form the second network, which can remain in the body for a longer period of time and facilitate the adhesion and growth of new chondrocytes. Together, the first network and the second network enable the complete self-assembly and supramolecular response function of DN hydrogels, thus, achieving the clinical goal of rapid self-healing, which is its advantage as a next-generation nanomaterial [62].

#### 4.4.2. Bioprinting Technology

In cartilage tissue engineering, the main advantage of 3D bioprinting is its ability to print scaffolds distributed with controlled cells that can promote cartilage tissue regeneration [149]. Thus, 3D bioprinting, as a precise and efficient biomanufacturing method for cartilage regeneration, can combine cells and biomaterials in an orderly manner while performing layer-by-layer deposition to precisely construct cartilage tissue scaffolds with a specific spatial structure [150]. The main raw materials for 3D bioprinting are organic nanomaterials and biological cells, which are layered to produce synthetic materials that can be implanted in cartilage defect sites. Compared to the traditional synthesis of organic nanomaterials, 3D bioprinting technology is faster and more concise in the way it synthesizes materials, facilitating the rapid application of new materials in cartilage repair [151]. The base materials for 3D bioprinting are organic nanomaterials. Take PEG, mentioned before, as an example, PEG is one of the most commonly used synthetic materials for 3D bioprinting [152]. In traditional production, PEG is more often used alone to make biological scaffolds [153]; now Yamasaki et al. [154] have mixed dimethacrylate-polyethylene glycol (PEG-DMA) material with human chondrocytes as raw material to print 3D scaffolds loaded with cartilage and found that the mechanical strength is much higher than that of traditional PEG scaffolds, and there is almost no effect on the biological activity of the cells, and the cell survival rate is much higher than that of traditional materials [155]. Grigor’eva et al. [156] proved that mixing methacrylic acid anhydride chitosan (PEG-GelMA) material with human mesenchymal stem cells for printing can promote the differentiation of stem cells, and the mechanical strength of the composite scaffold composed of both cells and organic nanomaterials breaks the bottleneck of the original single nanomaterial scaffold. In addition, PCL is also a commonly used bioprinting material. Similar to PEG, mixing PCL with traditional bionanomaterials results in scaffolds with higher resolution and significantly higher cell viability, as well as good thermoplasticity, mechanical strength, degradability, and biocompatibility [157]. The human-derived cells used in 3D bioprinting also have special properties. As shown in Figure 3, using chondrocytes as an example, Chen et al. [158] used stereolithography (SLA) technology to 3D print cartilage scaffolds. The scaffold significantly promoted the migration of chondrocytes in the cartilage defect area, and exosomes were released during the migration process, and exosomes were finally absorbed by chondrocytes. After the administration of mitochondria-related proteins, scaffolds can promote the generation of mitochondria in chondrocytes and restore some functions of chondrocytes that have been damaged, so as to realize the regeneration of cartilage defects. Along with the advancement of cell biology, the cells used in 3D bioprinting are more often mesenchymal stem cells and embryonic stem cells/induced pluripotent stem cells. The advantage of stem cells over traditional autologous chondrocytes is that they have good proliferative potential and do not lose their multidirectional differentiation ability within a few generations [159]. Ni et al. [160] demonstrated that scaffolds printed with a mixture of stem cells and organic nanomaterials can simultaneously promote the formation of new cartilage, integration of bone and cartilage, and lumen formation with significantly higher efficiency than using chondrocytes, and their safety has been demonstrated.

## 5. Conclusions and Outlook

The more structural layers of cartilage and the fact that articular cartilage often needs to bear more weight place higher demands on the materials needed for tissue engineering of articular cartilage. Organic nanomaterials have common features, such as higher mechanical strength, good biocompatibility, and induce chondrocyte secretion, which makes them commonly used materials in cartilage tissue engineering. Single or composite organic nanomaterials composed of different macromolecules have their own specific property preferences, which should be adjusted according to actual clinical needs. Advances in materials science and engineering have led to the rapid diversification of combinations between different materials, for example, composites composed of organic nanomaterials with cells and growth factors are beginning to gain importance. Such composites have unique advantages, such as low dosage, diverse functions, and good safety, and are expected to become the next generation of materials commonly used in tissue engineering. In addition, the multiple properties of bilayer network hydrogels meet the requirements of next-generation tissue engineering, i.e., diverse functions with less dosage, and their main function is to provide mechanical support for the composites and facilitate the generation of new cartilage tissue. In the future, the focus of cartilage repair research will be on natural cartilage signaling pathways, and clinical differentiation will be made for cartilage defects caused by different types of signaling pathway blockage, i.e., using high bioaffinity scaffolds to supplement the corresponding signaling pathway proteins, and implanting cartilage stem cells to achieve faster and better repair of cartilage damage.

## 6. Perspectives

With the development of materials science and engineering, as well as the analysis of cartilage repair signaling pathways, the treatment of cartilage repair in future clinical fields will tend to be precision medicine. For cartilage defects caused by diseases, doctors will first analyze the loss of signaling pathways in cartilage cells by means of genetic testing and proteomics, and then implant a high bioaffinity scaffold prepared by 3D printing with PEG as the main framework and filled with hyaluronic acid, and encapsulate cartilage stem cells with complete signaling pathways in the scaffold after repair. This treatment method can greatly reduce the rejection reaction and accelerate the speed of cartilage repair; for cartilage defects caused by sports injuries, a composite hydrogel prepared with hyaluronic acid and anti-inflammatory factors is used to protect the cartilage defect area from inflammatory factors, and then a three-dimensional scaffold prepared with PVA and chitosan is implanted to provide high mechanical strength to accelerate the cartilage repair.

## Figures and Tables

**Figure 1 bioengineering-09-00390-f001:**
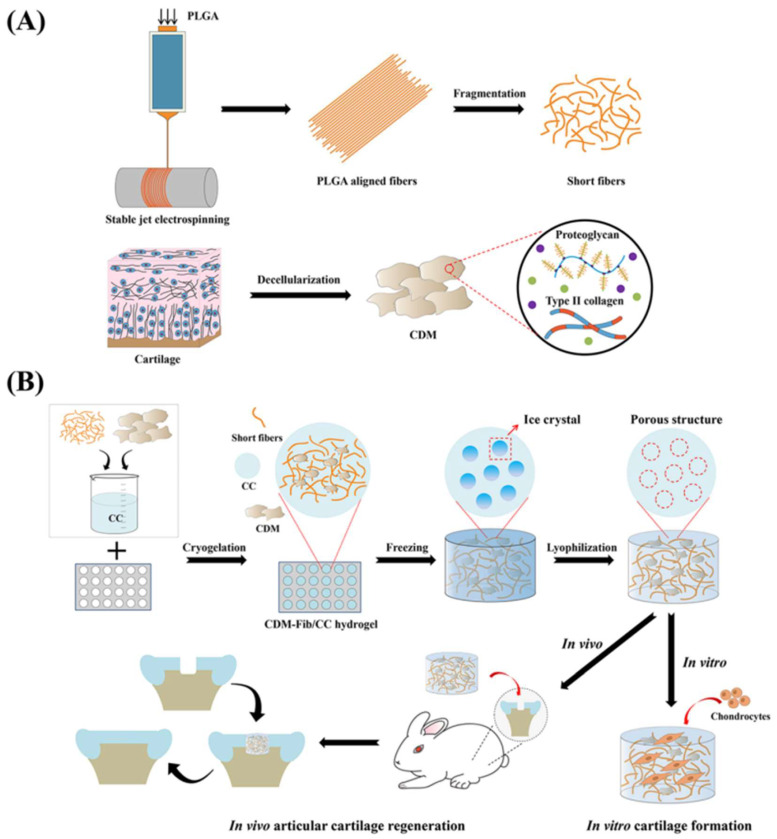
Shen et al., constructed a highly biomimetic PLGA-chitosan scaffold for cartilage regeneration. (**A**) Preparation of PLGA short fibers and CDM and (**B**) preparation of the PLGA-chitosan scaffold (CDM-Fib/CC). Reprinted with permission from Ref. [56]. 2021, American Chemical Society.

**Figure 2 bioengineering-09-00390-f002:**
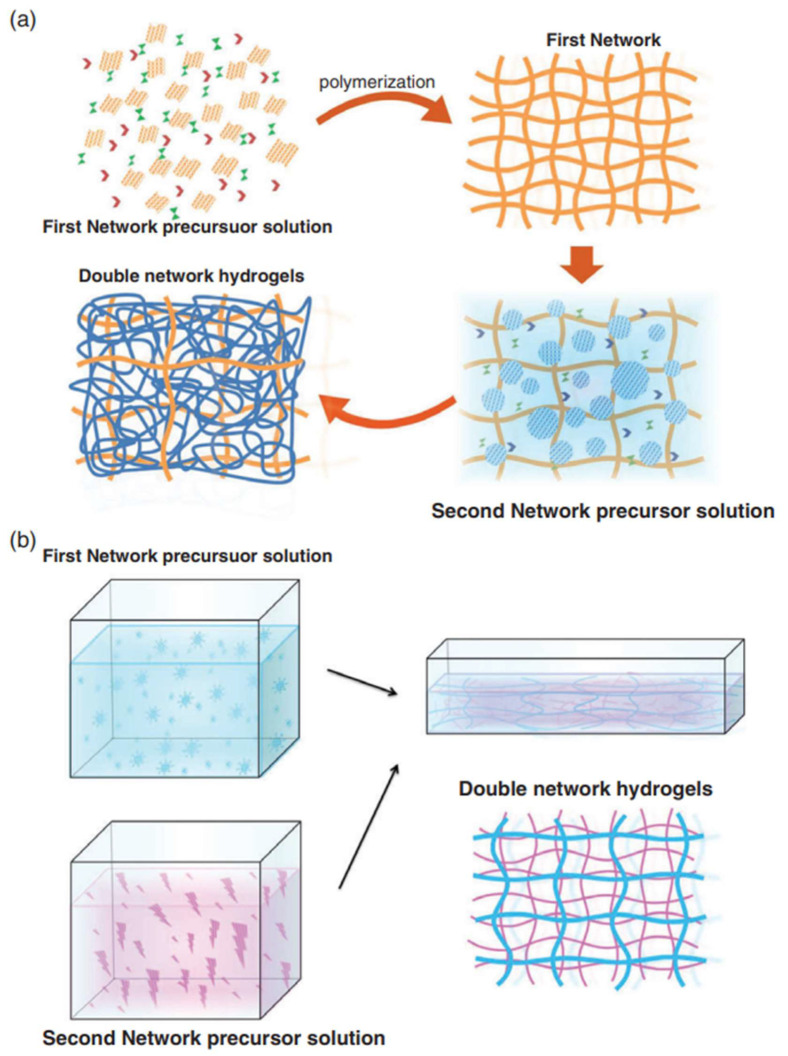
Schematic illustration of the preparation of DN hydrogels by chemically-chemically crosslinking. (**a**) Two-step polymerization method and (**b**) molecular stent method. Reprinted with permission from Ref. [145]. 2018, Wiley Periodicals, Inc.

**Figure 3 bioengineering-09-00390-f003:**
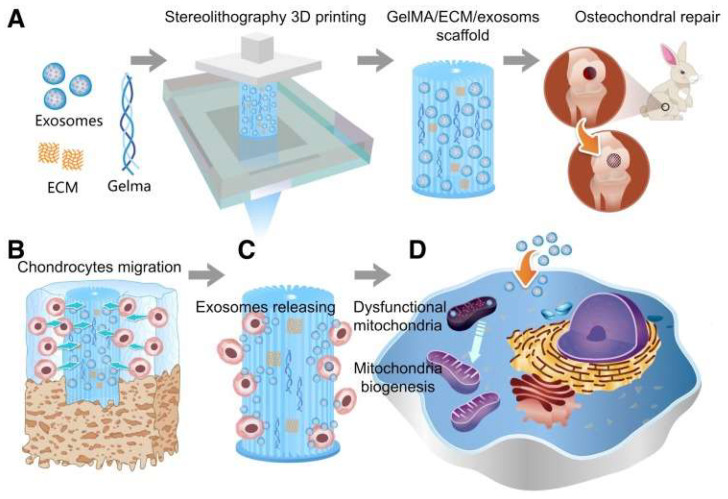
Schematic illustration of the one-step operation system for facilitating osteochondral defect regeneration. (**A**) Stereolithography-based bioprinting and osteochondral defect implantation. (**B**) Migration of chondrocytes to the defect regions. (**C**) Controlled administration of exosomes by the 3D printed scaffolds. (**D**) Enhanced chondrocyte mitochondrial biogenesis by the scaffolds [158].

## Data Availability

Not applicable.

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
