# Peer review of "Recent Developments and Current Applications of Organic Nanomaterials in Cartilage Repair"

_bioengineering, 2022, doi:10.3390/bioengineering9080390_

Round 1
Reviewer 1 Report
In this manuscript, the authors reviewed recent progresses and applications of organic nanomaterials in cartilage repair. The topic is interesting and fits well with Bioengineering journal. The manuscript was well written with a clear flow. Recent research work in the field were covered in the manuscript. Insights and perspective from the authors were provided. The publication of this manuscript should benefit researchers in the fields of nanomaterials and cartilage repair. Overall, it's a high quality review article. The reviewer would like to recommend the publication of it in Bioengineering.
Author Response
We appreciate your willingness to review this manuscript. We are also honored to receive your praise. Wish you a happy life!
Reviewer 2 Report
Three people as first author is a little too many!
It is a very descriptive review, useful in the educational/informative context
Figure 3 has to be better described in the text!
Conclusion is not clear and lines 585-589 have to be rewritten
Author Response
Please see the attachment.
Dear Reviewer 2,
Thank you very much for your valuable comments on our manuscript, bioengineering-1811618, entitled "Recent Developments and Current Applications of Organic Nanomaterials in Cartilage Repair". Your comments make us realize the inadequacy of our work and also bring us very important inspirations. Please allow me to report to you the details of our revision of this manuscript.
- Figure 3 has to be better described in the text!
Answer: We have added more details about Figure 3. Please see lines 555-561 for details.
- Conclusion is not clear and lines 585-589 have to be rewritten
Answer: We have rewritten this part. Please see lines 593-597 for details.
We hope you are satisfied with our revision. We are very grateful for your help!
With kind regards,
Xisheng Weng, MD
Department of Orthopaedics, Peking Union Medical College Hospital, Chinese Academy of Medical Sciences & Peking Union Medical College, Beijing 100730, China
E-mail: drwengxsh@163.com

Reviewer 3 Report
The submitted review reports an overview of the nanomaterials employed for the regeneration cartilage tissue damaged due to a trauma or a disease. Authors describe the performance of organic, composite and hydrogel-type materials used for cartilage regeneration. The manuscript includes a large number of references, namely 165. While the material part seems to be well structured and written the part related to the conclusion and future perspective is too generally approached. There is an imperious need for the authors to drawn clear and consistent conclusions from the study. Which is the best material, from those presented within manuscript, that is fit for cartilage regeneration? Which are the perspectives for these materials in order to reach clinical trials?
1. The Conclusion and outlook are too generally approached. There is need of consistence. Please provide in the last chapter “Conclusion and Outlook” or include a new chapter (“Perspectives”) in which the authors identify from the waste bibliography that they have analyzed, which is /are the best solution(s)/material(s) available in literature in order to reach clinical trials for cartilage regeneration. Which are the authors recommendations?
2. Please explain the significance of MSC within the manuscript text (line 138).
3. In the first two paragraphs of the chapter 4 entitled Organic nanomaterials for cartilage repair, the authors talk about the nano-inorganic materials and provide one example of CNTs for cartilage repair. Then the authors explained the advantages of the nano-organic materials over nano-inorganic ones. Why the chapter title is referred only to Organic nanomaterials for cartilage repair?
Author Response
Please see the attachment.
Dear Reviewer 3,
Thank you very much for your valuable comments on our manuscript, bioengineering-1811618, entitled "Recent Developments and Current Applications of Organic Nanomaterials in Cartilage Repair". Your comments make us realize the inadequacy of our work and also bring us very important inspirations. Please allow me to report to you the details of our revision of this manuscript.
- The Conclusion and outlook are too generally approached. There is need of consistence. Please provide in the last chapter “Conclusion and Outlook” or include a new chapter (“Perspectives”) in which the authors identify from the waste bibliography that they have analyzed, which is /are the best solution(s)/material(s) available in literature in order to reach clinical trials for cartilage regeneration. Which are the authors recommendations?
Answer: We have added a new chapter (“Perspectives”). Please see lines 603-616 for details.
- Please explain the significance of MSC within the manuscript text (line 138).
Answer: We have explained this word. Please see line 132 for details.
- In the first two paragraphs of the chapter 4 entitled Organic nanomaterials for cartilage repair, the authors talk about the nano-inorganic materials and provide one example of CNTs for cartilage repair. Then the authors explained the advantages of the nano-organic materials over nano-inorganic ones. Why the chapter title is referred only to Organic nanomaterials for cartilage repair?
Answer: We have changed the title of chapter 4. Please see line 140 for details.
We hope you are satisfied with our revision. We are very grateful for your help!
With kind regards,
Xisheng Weng, MD
Department of Orthopaedics, Peking Union Medical College Hospital, Chinese Academy of Medical Sciences & Peking Union Medical College, Beijing 100730, China
E-mail: drwengxsh@163.com
